# Ultra-High Frequency Ultrasound Imaging of Bowel Wall in Hirschsprung’s Disease—Correlation and Agreement Analyses of Histoanatomy

**DOI:** 10.3390/diagnostics13081388

**Published:** 2023-04-11

**Authors:** Tebin Hawez, Christina Graneli, Tobias Erlöv, Emilia Gottberg, Rodrigo Munoz Mitev, Kristine Hagelsteen, Maria Evertsson, Tomas Jansson, Magnus Cinthio, Pernilla Stenström

**Affiliations:** 1Department of Pediatric Surgery, Children’s Hospital, Skåne University Hospital Lund, Lund University, 221 85 Lund, Sweden; 2Department of Biomedical Engineering, The Faculty of Engineering, Lund University, 223 63 Lund, Sweden; 3Department of Clinical Genetics and Pathology, Skåne University Hospital, Lund University, 222 42 Lund, Sweden; 4Department of Biomedical Engineering, Skåne University Hospital Lund, The Faculty of Engineering, Lund University, 221 00 Lund, Sweden

**Keywords:** Hirschsprung’s disease, bowel wall, ultra-high frequency ultrasound, histopathology, children

## Abstract

Hirschsprung’s disease (HD) is characterized by aganglionosis in the bowel wall, requiring resection. Ultra-high frequency ultrasound (UHFUS) imaging of the bowel wall has been suggested to be an instantaneous method of deciding resection length. The aim of this study was to validate UHFUS imaging of the bowel wall in children with HD by exploring the correlation and systematic differences between UHFUS and histopathology. Resected fresh bowel specimens of children 0–1 years old, operated on for rectosigmoid aganglionosis at a national HD center 2018–2021, were examined ex vivo with UHFUS center frequency 50 MHz. Aganglionosis and ganglionosis were confirmed by histopathological staining and immunohistochemistry. Histoanatomical layers of bowel wall in histopathological and UHFUS images, respectively, were outlined using MATLAB programs. Both histopathological and UHFUS images were available for 19 aganglionic and 18 ganglionic specimens. The thickness of muscularis interna correlated positively between histopathology and UHFUS in both aganglionosis (R = 0.651, *p* = 0.003) and ganglionosis (R = 0.534, *p* = 0.023). The muscularis interna was systematically thicker in histopathology than in UHFUS images in both aganglionosis (0.499 vs. 0.309 mm; *p* < 0.001) and ganglionosis (0.644 versus 0.556 mm; *p* = 0.003). Significant correlations and systematic differences between histopathological and UHFUS images support the hypothesis that UHFUS reproduces the histoanatomy of the bowel wall in HD accurately.

## 1. Introduction

Hirschsprung’s disease is a congenital bowel motility disorder characterized by aganglionosis in the bowel wall. Inhibiting relaxation, aganglionosis leads to a life-threatening functional obstruction [1,2,3]. Treatment of Hirschsprung’s disease is by surgery, including resection of the aganglionic and transition zone segments, followed by the establishment of bowel continuity [4,5]. Aganglionosis always stretches in an oral direction from the anus. In the majority of patients, only the rectosigmoid colon is affected, but aganglionosis can also extend over a longer distance [6]. When deciding upon the length of bowel to be resected, intraoperative fresh frozen biopsies are required to confirm the level of ganglionosis. Acknowledged clinical problems are that the frozen biopsy method is a rather blunt technique, and that resection of a too short or a too long bowel segment can cause severe postoperative problems [7,8]. Additionally, waiting times for frozen biopsy analyses during surgery can be considerable, up to several hours, especially if multiple biopsies are needed as a result of an unexpected aganglionic extension, thus burdening the child with having to endure prolonged anesthesia. A more precise and instant intraoperative diagnostic method is warranted to make Hirschsprung’s disease surgery quicker and safer. Ultra-high frequency ultrasonography (UHFUS) has been suggested to be such an instant method that could potentially replace frozen biopsy [9]. UHFUS with frequencies of <70 MHz captures superficial depths of a few millimeters with a resolution down to 30 μm [10]. This is compared to conventional ultrasound with 2–15 MHz frequencies used in clinical settings and imaging depths of 2–20 cm. UHFUS imaging has been reported to have diagnostic potential within the fields of dermatology, vascular medicine, musculoskeletal evaluation, and gastrointestinal surgery with regard to Hirschsprung’s disease [9,11]. In diagnostics of Hirschsprung’s disease, a pilot study showed that UHFUS can potentially be useful in differentiating between the aganglionic and ganglionic bowel wall [9]. This could be because histoanatomical landmarks have been shown to differ between ganglionosis and aganglionosis [9,12]. However, there remains a gap in knowledge regarding the accuracy of UHFUS in reproducing the histoanatomical layers of the bowel wall.

Therefore, the overall aim of this study was to explore whether the histoanatomical layers of the bowel wall could be imaged accurately by UHFUS. The first research question was to establish whether the thicknesses of the histoanatomical layers of the muscularis interna and muscularis externa, as seen on UHFUS images of fresh bowel wall ex vivo, correlate to those of the histopathologically-prepared specimen. The second question was to ascertain whether any systematic differences and low agreement of histoanatomical thicknesses were evident when comparing the bowel wall thicknesses as measured on histopathology and UHFUS images.

The first hypothesis was that morphometrics, i.e., thicknesses of histoanatomical layers as measured by UHFUS imaging and histopathological specimens, respectively, would correlate well. The second hypothesis was that systematic differences and a low agreement between morphometrics of bowel walls on histoanatomy and UHFUS would be evident as a result of histopathology preparation effects of specimens [13,14,15].

## 2. Materials and Methods

### 2.1. Patients

This study was a translational observational study performed at a national referral center for Hirschsprung’s disease covering a geographical uptake area of 5 million residents. Morphometrics of formalin-prepared hematoxylin–eosin-stained bowel wall specimens were compared with those of the same patient’s fresh bowel wall imaged on UHFUS ex vivo.

All children with Hirschsprung’s disease who were to undergo surgery with resection of the aganglionic bowel segment at the Department of Pediatric Surgery (a national center for Hirschsprung’s disease) from April 2018 to December 2021 were eligible for inclusion. Information about the patient’s age, weight, and length of resected bowel segments were retrieved from the local Hirschsprung’s disease register with data collected prospectively. The inclusion criterion for correlation analyses was rectosigmoid aganglionosis stretching 5–30 cm, according to the pathology report, in children weighing under 10 kg and being younger than 1 year of age. The surgical resection length of aganglionosis was decided based upon the pathologist’s analysis of intraoperative fresh frozen biopsies taken by the pediatric surgeon, confirming the presence of ganglionic bowel wall. The accuracy of the frozen biopsy results was confirmed by full histopathological analyses by histopathological staining with hematoxylin–eosin and immunohistochemistry (calretinin and S-100) [16,17,18] in a final pathology report of the whole resected specimen.

### 2.2. Specimen Treatments—Fresh and Histopathological

For UHFUS imaging, the Vevo MD ultrasound scanner (FUJIFILM VisualSonics Inc., Toronto, ON, Canada) equipped with a UHF70 transducer, delivering a center frequency of 50 MHz, was used. After surgical resection of the bowel, the retrieved fresh bowel specimen was pinned to a cork mat and examined ex vivo using UHFUS from the serosal (outer) surface (Figure 1). A gel layer was used as a conductor between the transducer and the bowel wall. Minimal pressure was applied to the bowel during the examination in order to avoid manipulation of the examined specimen. UHFUS images were taken longitudinally and saved as B-mode images. For each patient, UHFUS images of both aganglionic and ganglionic bowel walls were saved prospectively in a database. We used a predefined ultrasound acquisition protocol that included center frequency (50 MHz), power (100%), gain (48 dB), depth (7 mm), width (9.73 mm), persistence (off), and dynamic range (65 dB). The settings, including the depth-depending gain, could be optimized during scanning. Two pediatric surgeons (CG, PS) performed all the UHFUS examinations together. They established the working procedure together and assisted each other in imaging and sampling. The bowel specimen was thereafter treated with formalin and sent to the Department of Clinical Genetics and Pathology, where aganglionic and ganglionic segments were confirmed by microscopy after paraffin embedding and standard histopathological staining with hematoxylin–eosin and immunohistochemistry (calretinin and S-100). Histoanatomical images were saved cross-sectioned in the data system Laboratory Information Management System (LIMS) RS Pathology^®^. Images with hematoxylin–eosin staining were assessed. UHFUS and histopathological images of poor quality, as a result of damaged tissue, broken specimens, or air interfaces causing shadows in the UHFUS imaging, were excluded from the study.

### 2.3. Assessment and Measurements

Histopathological and UHFUS images were assessed using two in-house software programs based on MATLAB (MathWorks Inc., Natick, MA, USA). One program for histopathology and one for UHFUS, respectively, were developed within the research project. In the histopathological MATLAB program, the external and internal borders of the muscular layers were delineated manually for the muscularis externa and muscularis interna, respectively (Figure 2). The histopathology images could be assessed either whole or in parts. White areas within the tissue were automatically erased before calculations, as programmed. This was decided upon by the assessor in order to avoid sections with image or specimen artifacts. In the UHFUS images, a 5 mm long region of interest (ROI) in the MATLAB program was selected by the assessor and decided upon with respect to image quality and well-represented histoanatomy in the B-mode. The ROI was selected by the same pediatric surgeons who performed the UHFUS examinations. Within this ROI, the presumed muscularis externa and interna were delineated manually, in a similar manner to the procedure when taking the histopathology images. Muscular layer thicknesses were generated automatically from measurement intervals of every 14 µm in the histopathology images and 32 µm in the UHFUS images, respectively (Figure 2).

### 2.4. Statistics

Measurements of the muscularis externa and interna, generated from delineation in MATLAB, were given as mean thickness with standard deviation (SD), and median thickness with range (minimum to maximum). These calculations were performed for both aganglionosis and ganglionosis in histopathological and UHFUS images, respectively. Data management and statistical analyses were performed using Microsoft^®^ Excel and IBM^®^ SPSS^®^ statistics version 27. Ratios of the muscularis interna and externa thicknesses were calculated and used in systematic difference analyses (paired testing). Correlation between histopathological and UHFUS images, and their strength and direction, was assessed by the Spearman correlation test on a group level. For exploring systematic differences between the modalities, the Wilcoxon Signed Rank Test was used, in which the patient served as their own control. A *p*-value of <0.05 was considered to be statistically significant.

Agreement referred to whether histoanatomical thicknesses were close or differed between the histopathological specimen and UHFUS images. Agreement between thicknesses was visualized and interpreted by Bland–Altman plots: a method that has proven to be useful in comparing diagnostic modalities [19,20].

### 2.5. Ethical Considerations

Ethical approval was obtained from the local ethics review board (DNR 2017/769). Oral and written information was given, and the guardians’ written consents were obtained.

## 3. Results

### 3.1. Patient and Specimen Characteristics

During the study period, 36 children underwent surgery for Hirschsprung’s disease. According to the study criteria, 16 children were excluded (Figure 3). Two histopathological specimen images (one aganglionic and one ganglionic) and one UHFUS image (ganglionic) were excluded as a result of artifacts in the specimen and/or images. Thus, in total, 19 aganglionic and 18 ganglionic bowel specimens from 20 patients were included. The median age of the included patients was 29 days (range: 11–174 days) and their median weight was 4012 g (range: 2600–7700 g) at the time of surgery. The median length of the formalin-fixed resected bowel specimen was 17 cm (range 7–26 cm).

### 3.2. Aganglionic Bowel Wall

In aganglionosis, the thickness of the muscularis interna correlated positively between histopathology results and UHFUS in aganglionosis, i.e., the thicker the muscularis interna was in the histopathological specimen, the thicker it was on UHFUS imaging (R = 0.651, *p* = 0.003, Spearman correlation analyses) (Figure 4a and Appendix A). This correlation housed a systematic difference in the thickness of the muscularis interna, which was, on average, 0.168 mm thicker in histopathological images than on UHFUS (*p* < 0.001) (Table 1).

### 3.3. Ganglionic Bowel Wall

In ganglionic bowel wall specimens, the thickness of the muscularis interna also correlated positively between histopathology results and UHFUS (R = 0.534, *p* = 0.023) (Figure 4b and Appendix A). The correlation housed a systematic difference, being, on average, 0.136 mm thicker in histopathology images compared to images achieved using UHFUS (*p* = 0.003) (Table 1). In ganglionosis, the muscularis externa differed systematically, being thinner in histopathological specimens than on UHFUS images (*p* = 0.006) (Table 1). The ratio of the thickness of the muscularis interna/muscularis externa in ganglionosis was systematically greater in histopathological specimens than on UHFUS (*p* < 0.001) (Table 1).

### 3.4. Agreements

In line with the result of systematic differences, agreements between histoanatomical thicknesses of muscularis interna and externa in histopathological and UHFUS images were low (Figure 5).

## 4. Discussion

According to the study results, UHFUS visualized histoanatomical muscular layers of bowel walls in children with Hirschsprung’s disease accurately. The histoanatomical muscular layer thicknesses, and especially those of the muscularis interna, correlated well between UHFUS imaging and histopathology. As expected, as a result of speculated histopathological preparation effects, systematic differences and low agreement were evident between the UHFUS images and histopathology results. The study’s confirmation of a reliable reproduction by UHFUS of the thicknesses of the bowel wall’s histoanatomical muscular layers supports our previous report on the potential of UHFUS in imaging the bowel wall [9].

Studies aiming to correlate the histoanatomy of the bowel wall to findings on ultrasound have been carried out previously, but only by exploring the use of lower frequency ultrasound and never UHFUS. In these previous reports, the use of ultrasound of 8.5 and 25 MHz on human bowel did not visualize histological layer thicknesses convincingly [21,22]. The lack of correlation was attributed to the small histoanatomical sizes and the physical principles of ultrasound, i.e., how the soundwaves interacted differently with various tissues. These studies were limited by the use of low- and normo-resolution ultrasound, leading to less detailed images compared to those that can be visualized by UHFUS on small tissues over short distances [11]. Advantages of ultra-high resolution imaging over short distances, such as when visualizing bowel wall, and our here-reported UHFUS results, are supported by one animal study suggesting strong and positive correlations between histological and ultrasound morphometrics of both the muscularis interna and muscularis externa in bowel wall [23].

One factor to be taken into consideration is that ultrasound accuracy might depend upon the examiner’s handling and positioning of the transducer. This is also of concern in the validation of UHFUS. In the animal study referred to above [23], the transducer was positioned in water at 1 cm from the sample, in order to avoid any modifying pressure on the tissue. This meticulous methodology was not repeated in our study because we aimed for a clinically feasible setting, implying an almost direct contact between the transducer and bowel. Therefore, we cannot exclude the possibility that pressing the transducer on the bowel wall might have influenced the muscular thicknesses in the UHFUS images. This could have affected the muscularis externa in particular, which is the layer closest to the transducer and could, therefore, be most influenced by external pressure. However, the external pressure imposed by the transducer will also be the case in the clinic, requiring consideration in our forthcoming further methodological validation and refining work. The individual learning curve can influence the accuracy and reliability of the detailed UHFUS, and this needs to be taken into consideration and addressed before diagnostics can be implemented in clinical work.

Another factor to be considered, both in the validation process and in the clinic, is the user-dependent quality of imaging and subjective interpretation of images. In the present study, two surgeons performed the UHFUS assessments, and interpretations were confirmed using MATLAB programming. Nevertheless, before clinical implementation of the UHFUS technique can occur, interpersonal variability testing of the UHFUS on the bowel will be required as part of the validation process.

As expected, significant systematic differences were observed. These were hypothesized to be a result of the expected effects of histopathological preparation. Our finding that the muscularis interna was systematically thicker on histopathological imaging than on UHFUS was in line with the results of a preclinical animal model study suggesting a swelling of muscular tissue post-formalin fixation [24]. Notably, there are also studies suggesting a shrinkage of the bowel due to formalin fixation [13,14,15]. However, in contrast to our study, those bowel specimens were analyzed regarding total specimen length, while analyses in our study were cross-sectional, focusing on separate muscularis layers. Speculating, one explanation for the shrinking versus swelling of the tissue could be the direction of the assessment. This is because a whole bowel specimen comprises a mix of several histoanatomical layers, and their various fiber directions might lead to a total shrinking effect in length. Still, within the whole specimen, separate histoanatomical layers might swell, which could possibly be identified when the specimen is studied cross-sectionally.

A strength of our study was the fairly homogenous cohort, including only children with recto-sigmoid aganglionosis who were of similar ages and weights. The majority of specimens and UHFUS images were generally of high quality, containing well-preserved histoanatomical bowel wall layers. Although limited, a need for the exclusion of samples as a result of low image quality was evident in three of 40 cases. Image quality requires full consideration in the clinic because safe diagnostics must be secured for all patients. Greater experience, methodological refinements, and technical improvements are expected to contribute to an improved image quality. Specific strategies for improving image quality could be to repeat imaging in every patient, in order to secure the highest quality of results.

Another strength of the study was that both aganglionic and ganglionic areas were pinpointed specifically on both fresh and fixated bowel samples and, therefore, allowed correlation of exactly the same areas. This also enabled the patient to serve as their own control in paired statistical tests, minimizing effects by confounding factors, such as increased bowel wall thickness with age [12,25]. Furthermore, the two MATLAB programs, developed specifically for analyses of UHFUS and histopathological imaging, enabled precise delineations of the histoanatomical muscle layers and the programs supported the generation of outcomes objectively. These MATLAB programs enable the unique, novel, and exact technique to assess the histoanatomy of the bowel wall, which will be useful in future studies.

One obvious limitation of the study was the sparse number of specimens which diminished the probability of revealing the absence of differences. In addition to pressure effects, there were only a few patients who might have contributed inconsistent findings of aganglionic muscularis externa. Additionally, as a result of the great diversity in thicknesses of the muscularis externa, particularly in aganglionosis, the statistical power was low. Moreover, the UHFUS examinations were not standardized with regard to the transducer angle or gel amount, which might have impacted the measured layer thicknesses of the bowel wall. Additionally, histopathological images were cross-sectioned while UHFUS images were sectioned longitudinally. Since thicknesses might appear differently if cross- or longitudinally sectioned, the direction of the transducer might have influenced the outcome. For statistical limitations, assessments using MATLAB programming of both histopathology and UHFUS were performed by only one person, which might have skewed data. Similarly, the Bland–Altman agreement analyses were, by their nature, linked to a subjective interpretation; however, this potential bias was minimized by having a statistician assist in the interpretation of the results. For an intervariability control, analyses between UHFUS users will be analyzed separately (cohen kappa) in a next step. Then, for transparency, objectivity, and to collect more data, a multicenter study for validation of the use of UHFUS in the diagnostics of bowel wall pathology is warranted. This is now planned, in the form of a long-term outcome report.

This study is part of a larger project aiming to develop and validate the use of UHFUS in Hirschsprung’s disease, as a novel and immediate method to delineate between aganglionic and ganglionic bowel. Adding knowledge about the validity of UHFUS in the histoanatomical preciseness of the bowel wall, this study serves as an important and essential start for more studies on UHFUS in the diagnostics of Hirschsprung’s disease.

## 5. Conclusions

This study reveals that UHFUS imaging replicates the histoanatomy of the bowel wall adequately. The thicknesses of the histoanatomical muscular layers, and especially the muscularis interna, correlated well between histopathological and UHFUS images, and the expected systematic differences as a result of histopathological preparations were confirmed. This study serves as a profound base for the use of UHFUS as a novel method in the diagnostics of Hirschsprung’s disease.

## Figures and Tables

**Figure 1 diagnostics-13-01388-f001:**
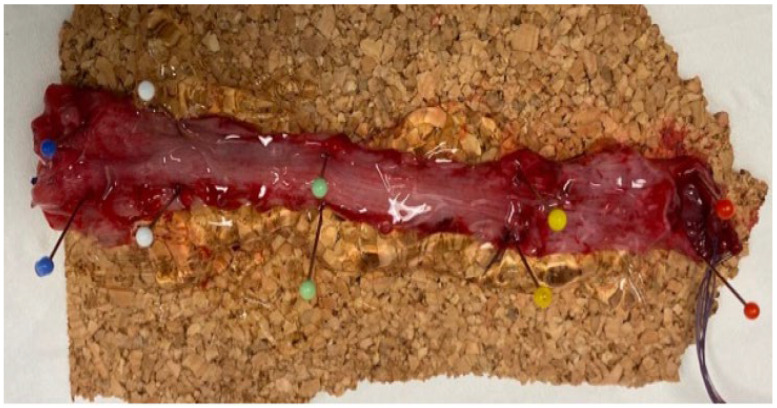
Resected bowel of a child with Hirschsprung’s disease. Pinpointed aganglionic bowel (blue pins) and ganglionic bowel (yellow pins) represent the sites where the histoanatomical thicknesses on ultra-high frequency ultrasound (UHFUS) (ex vivo) and histopathology were correlated and compared.

**Figure 2 diagnostics-13-01388-f002:**
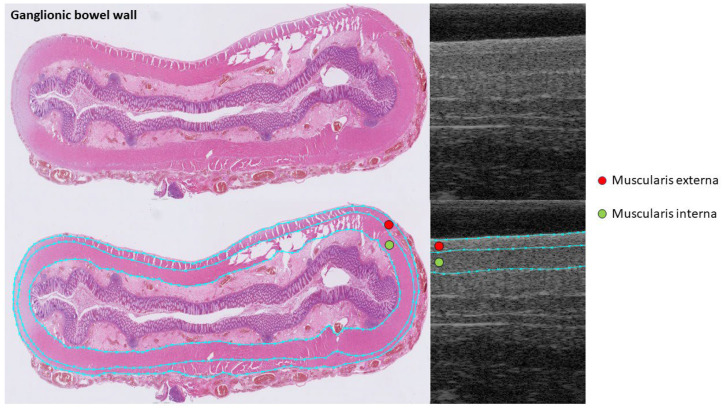
Histoanatomical and ultra-high frequency ultrasound (UHFUS) images of the ganglionic bowel wall of the same patient. The histopathological specimens (**left**) were stained with hematoxylin–eosin. UHFUS imaging (**right**) was performed on bowel ex vivo with center frequency 50 MHz. The method of how to delineate muscularis externa’s and interna’s outermost limits using MATLAB software is shown in the lower images.

**Figure 3 diagnostics-13-01388-f003:**
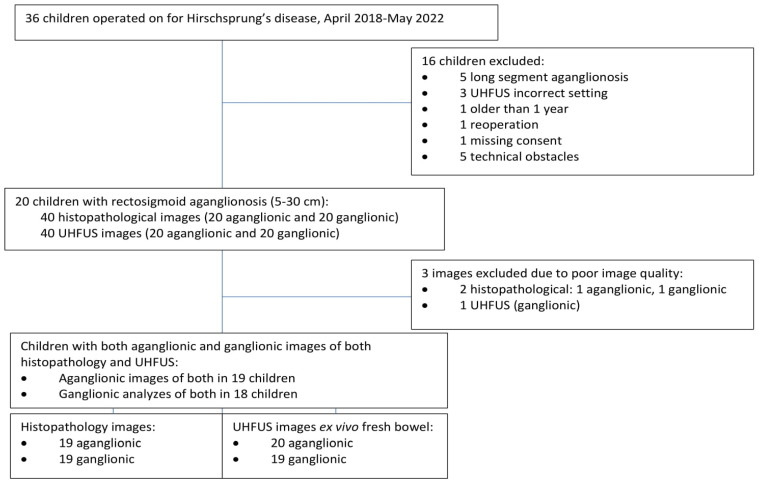
Flow chart of patients and images included in the study. UHFUS: ultra-high frequency ultrasound.

**Figure 4 diagnostics-13-01388-f004:**
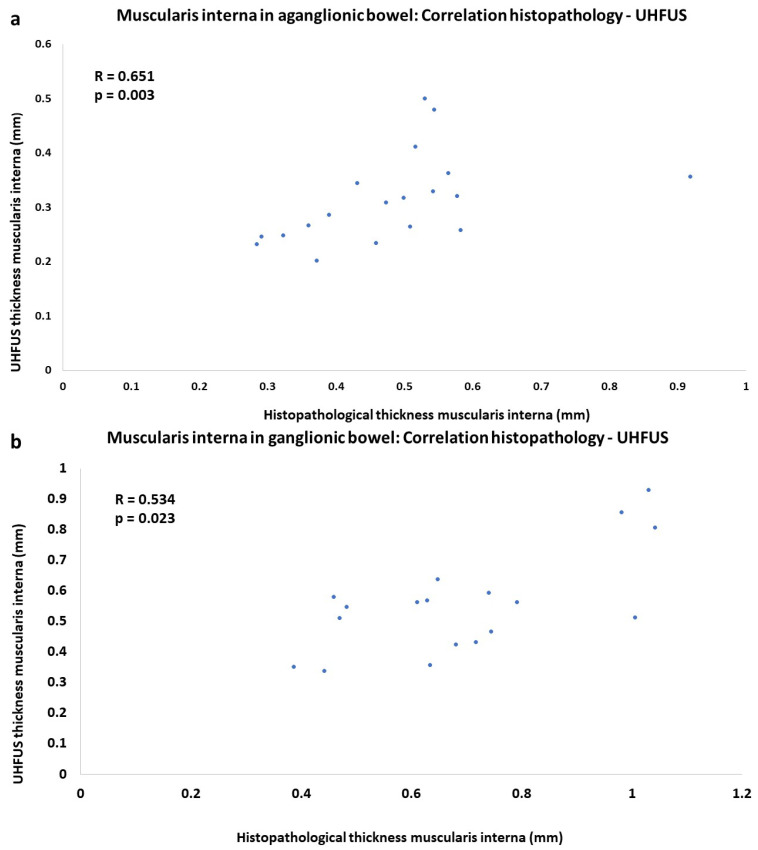
Correlation between muscularis interna thickness in histologically prepared specimens and ultra-high frequency ultrasound (UHFUS) images of (**a**) aganglionic bowel wall, *n* = 19, and (**b**) ganglionic bowel wall, *n* = 18. Spearman rank correlation coefficient (R).

**Figure 5 diagnostics-13-01388-f005:**
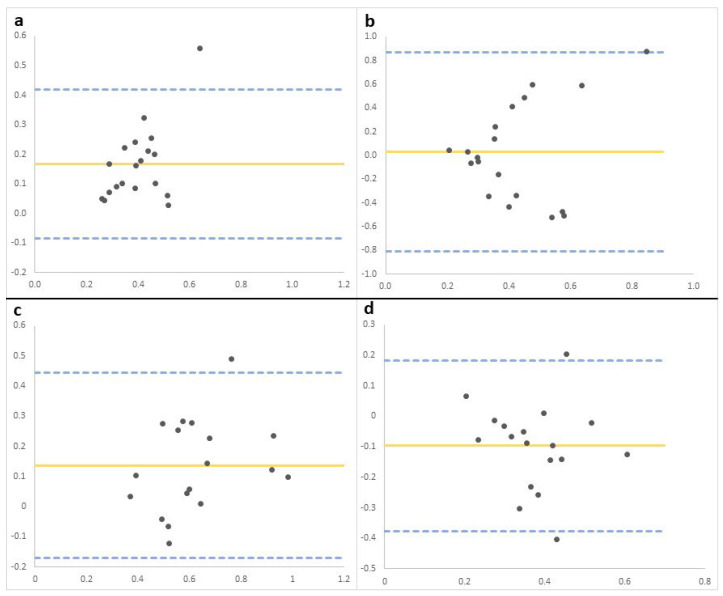
Agreement scatterplots (Bland–Altman) showing low agreement (uneven distribution) of thicknesses (mm) of histoanatomical layers as measured in histopathological and UHFUS images. The X-axis shows the mean thickness of the histoanatomical layer of histopathology and UHFUS in the same patient. The Y-axis shows the mean difference of thickness between the histoanatomical and UHFUS layer thickness in the same patient. The yellow horizontal lines represent degrees of agreement. The blue dashed horizontal lines represent ± 2 SD of histoanatomical thickness difference. Aganglionosis: Agreement of thicknesses of (**a**) muscularis interna and (**b**) muscularis externa, *n* = 19. Ganglionosis: Agreement of thicknesses of (**c**) muscularis interna and (**d**) muscularis externa. *n* = 18.

**Table 1 diagnostics-13-01388-t001:** Systematic differences of histoanatomical dimensions in bowel wall comparing histopathologically-prepared bowel specimens and ultra-high frequency ultrasound (UHFUS) of ex vivo examined bowel samples. Thicknesses were assessed using MATLAB programs. Median was calculated as the 50th percentile of individual mean measurements of histoanatomical thicknesses.

	Thickness in Aganglionosis *n* = 19			Thickness in Ganglionosis *n* = 18		
Histoanatomical Layer	Histopathology (mm) Median (Range)	UHFUS Image (mm) Median (Range)	Systematic Difference *p*-Value ^1^	Histopathological Bowel Wall Specimen (mm) Median (Range)	UHFUS Image (mm) Median (Range)	Systematic Difference *p*-Value ^1^
Muscularis interna (mm)	0.499 (0.284–0.918)	0.309 (0.202–0.500)	**<0.001**	0.664 (0.386–1.042)	0.556 (0.338–0.931)	**0.003**
Muscularis externa (mm)	0.291 (0.165–1.285)	0.322 (0.175–0.830)	0.872	0.297 (0.186–0.556)	0.433 (0.169–0.668)	**0.006**
Ratio: muscularis interna/ muscularis externa	1.253 (0.492–2.257)	0.888 (0.382–2.074)	0.064	2.101 (1.290–3.247)	1.333 (0.723–2.059)	**<0.001**

^1^ Related-Samples Wilcoxon Signed Rank Test.

## Data Availability

The data presented in this study are available on reasonable request from the corresponding author.

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
