# Peer review of "Ultra-High Frequency Ultrasound Imaging of Bowel Wall in Hirschsprung’s Disease—Correlation and Agreement Analyses of Histoanatomy"

_diagnostics, 2023, doi:10.3390/diagnostics13081388_

Round 1

Reviewer 1 Report

Dear Authors

This is an interesting paper.

I would like the Authors to highlight the learning curve needed to be confident in diagnosing HD with US. It would be also interesting to see if a multicentre study is potentially needed. This is an important point as could help many colleagues that don't have pathologists on site.

It is important to have long terms outcomes in a follow-up paper.

Reviewer 2 Report

The manuscript addresses the relevant issue of bowel wall evaluation in Hirschsprung’s disease. There are few minor issues requiring authors' attention:

- UHFUS imaging section should report the acquisition protocol employed, including technical parameters (gain, TGC, etc..)

- Please report how calibration between the two paediatric surgeons was performed (prior to study beginning on samples not included in the study?), and the Cohen kappa value.

- Please specify how the ROI was selected on UHFUS images.
